# Upgrade of the CMS Barrel Electromagnetic Calorimeter for the High Luminosity LHC

**Charlotte Cooke** [1,2] **on behalf of the CMS Collaboration**

1 School of Physics, University of Bristol, Bristol BS8 1TH, UK; charlotte.cooke@bristol.ac.uk
2 Rutherford Appleton Laboratory, Didcot OX11 0QX, UK

**Abstract:** The high luminosity upgrade of the LHC (HL-LHC) at CERN will provide unprecedented instantaneous and integrated luminosities of up to $7.5 \times 10^{34}$ cm$^{-2}$s$^{-1}$ and 4500 fb$^{-1}$, respectively, from 2029 onwards. To cope with the extreme conditions of up to 200 collisions per bunch crossing, and increased data rates, the on- and off-detector electronics of the CMS electromagnetic calorimeter (ECAL) will be replaced. A dual gain trans-impedance amplifier and an ASIC providing two 160 MHz ADC channels, gain selection, and data compression will be used. The lead tungstate crystals and avalanche photodiodes (APDs) in the current ECAL will keep performing well and will therefore be maintained. The noise increase in the APDs, due to radiation-induced dark currents, will be minimised by reducing the ECAL operating temperature from 18 °C to around 9 °C. Prototype HL-LHC electronics have been tested and have shown promising results. In two test beam periods using the CERN SPS H4 beamline and an electron beam, the new electronics achieved the target energy resolution and a timing resolution consistent that is consistent with our requirements of 30 ps timing for energies greater than 50 GeV.

**Keywords:** electromagnetic calorimeter; compact muon solenoid; high luminosity large hadron collider

## 1. Introduction

The CMS electromagnetic calorimeter (ECAL) is a lead tungstate (PbWO$_4$) crystal calorimeter which provides excellent energy resolution in the harsh radiation environment of the LHC [1]. Specifically, it has achieved a 1% mass resolution for the Higgs Boson in the $\gamma\gamma$ decay channel [2]. It is a hermetic and compact detector with coverage up to $|\eta| = 3$, split into barrel and endcap regions as shown in Figure 1. The barrel region contains 61,200 crystals across 36 supermodules and uses avalanche photodiodes (APDs) as photodetectors. The endcap region contains 14,648 crystals in four half disk dees and uses vacuum phototriodes. The barrel covers the region $|\eta| < 1.48$ while the endcap covers $1.48 < |\eta| < 3$.

The main objective of the high luminosity LHC (HL-LHC) is to deliver a much larger dataset for physics analysis to the LHC experiments. The CMS ECAL will have to maintain physics performance for an integrated luminosity of 4500 fb$^{-1}$, a peak luminosity of $7.5 \times 10^{34}$ cm$^{-2}$s$^{-1}$ and for 200 pileup interactions, which is the number of simultaneous proton–proton collisions per LHC bunch crossing. In Run 3 (the current data taking period), CMS is expected to reach an integrated luminosity of 30 fb$^{-1}$, a peak luminosity of $2.2 \times 10^{34}$ cm$^{-2}$s$^{-1}$ and up around 60 pileup. This has several implications. Firstly, the trigger latency will be increased from 4 µs to 12.5 µs, which will allow the use of tracks in the Level 1 (L1) trigger [3]. The L1 trigger rate will also be increased by a factor of 7.5, going from 100 kHz to 750 kHz, with a high level trigger (HLT) rate of up to 10 kHz. Event reconstruction will be more challenging with much higher in-time (due to collisions in the same bunch crossing) and out-of-time pileup (due to collisions in different bunch crossings). The detector will also be subject to more radiation, leading to more noise in the form of increased APD leakage current and crystal transparency loss. There will also be an

increased rate of anomalous signals ("spikes"), which are caused by hadrons impacting directly on the APDs.

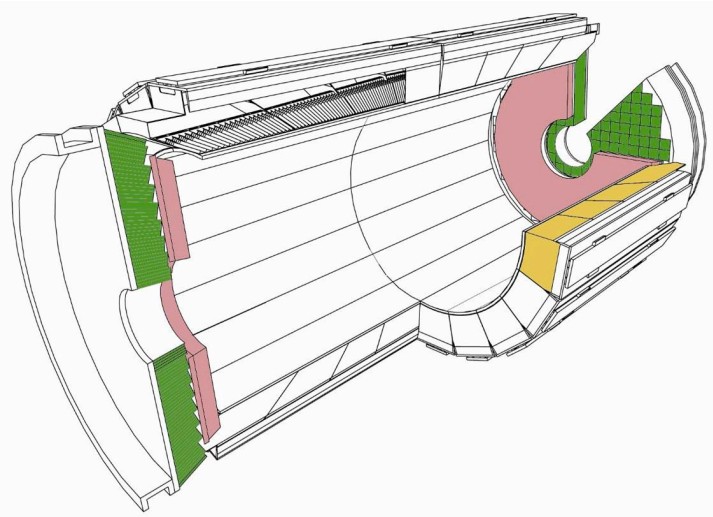

**Figure 1.** Layout of the CMS ECAL. One of the 36 barrel supermodules is highlighted in yellow, and the endcaps are highlighted in green.

## 2. Materials and Methods

The current ECAL on-detector electronics is comprised of a very front end (VFE) and front end (FE). The VFE card, which serves five readout channels, contains multi-gain pre-amplifiers (MPGAs), charge sensitive amplifiers with 3 output gain values: $\times 1$, $\times 6$ and $\times 12$, and multi-channel ADCs which have a resolution of 12 bit/gain value and a sampling frequency of 40 M samples per second. The FE card, which receives data from up to five VFE cards, controls the data pipeline and trigger primitive generation, which is information used by the L1 trigger processors. In this system the trigger data granularity is an array of $5 \times 5$ crystals.

The plan for the HL-LHC is to replace the endcaps with a high granularity calorimeter because of the radiation damage that will be sustained, particularly at high $|\eta|$ [4]. All ECAL barrel supermodules will be refurbished during long shutdown 3 (LS3), between 2026 and 2028. The lead tungstate crystals and APDs in the barrel will be retained, though the operating temperature will be reduced from 18 $^\circ$C to 9 $^\circ$C to keep noise levels below 250 MeV. This new operating temperature requires new coolant distribution pipes carrying chilled water at 6 $^\circ$C, though the current cooling distribution inside the supermodule will not require any modifications. Further cooling would require an extra chiller with significantly larger capacity. The on-detector electronics will be replaced with new radiation hard ASICs with faster pulse shaping and a factor of 4 increase in the sampling rate. A schematic detailing the HL-LHC electronics can be seen in Figure 2. This will reduce impact of out-of-time pileup and limit the increase in APD noise, as well as improve spike rejection at L1 via pulse shape discrimination. For energies greater than 50 GeV, 30 ps timing resolution will be achieved.

A new streaming front-end board will provide single crystal information to the L1 trigger via high speed radiation hard optical links (lpGBT). The new off-detector board will allow the use of more advanced algorithms in high performance FPGAs.

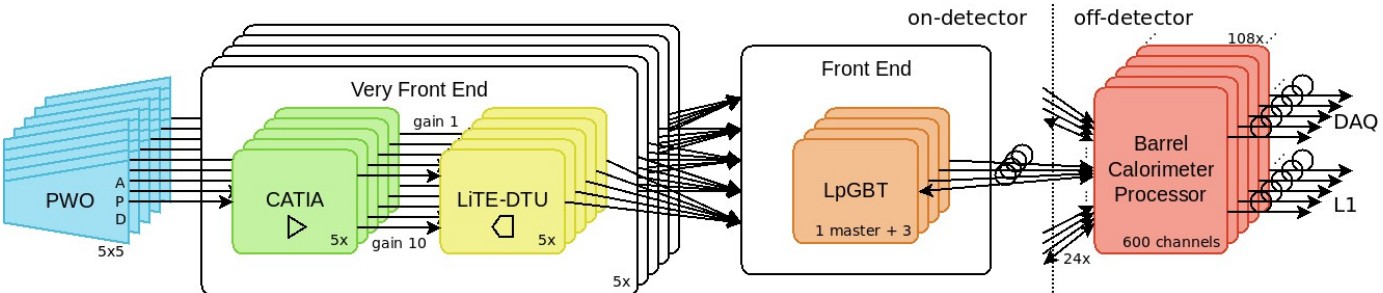

**Figure 2.** Schematic showing the layout for the CMS ECAL on- and off-detector readout for HL-LHC for a 5 × 5 crystal matrix.

## 3. Results

### 3.1. Lead Tungstate Crystal Longevity

The main concern for the ECAL crystals in the HL-LHC is ageing due to radiation damage. The scintillation mechanism is not affected by radiation, however radiation reduces the crystal transparency and therefore light output. This effect is monitored and corrected for using a dedicated light injection system. The change in light output between 2011 and 2018 can be seen in Figure 3. The relative response to laser light decreases with time due to radiation damage. This reduction is most pronounced at high $\eta$, while the ECAL barrel still retains ~90% of its response. Monte Carlo simulations have been used to predict the light output in the HL-LHC era and these have been validated using test beam data studying the effect of hadron irradiation on crystal response. These results can be seen in Figure 4. The target integrated luminosity of 4500 fb$^{-1}$ at the end of the HL-LHC would lead to a relative light yield between 25% and 45% depending on $\eta$. This is comparable to the current operating conditions in the ECAL endcap, giving confidence that we will be able to operate the barrel successfully with this level of light output.

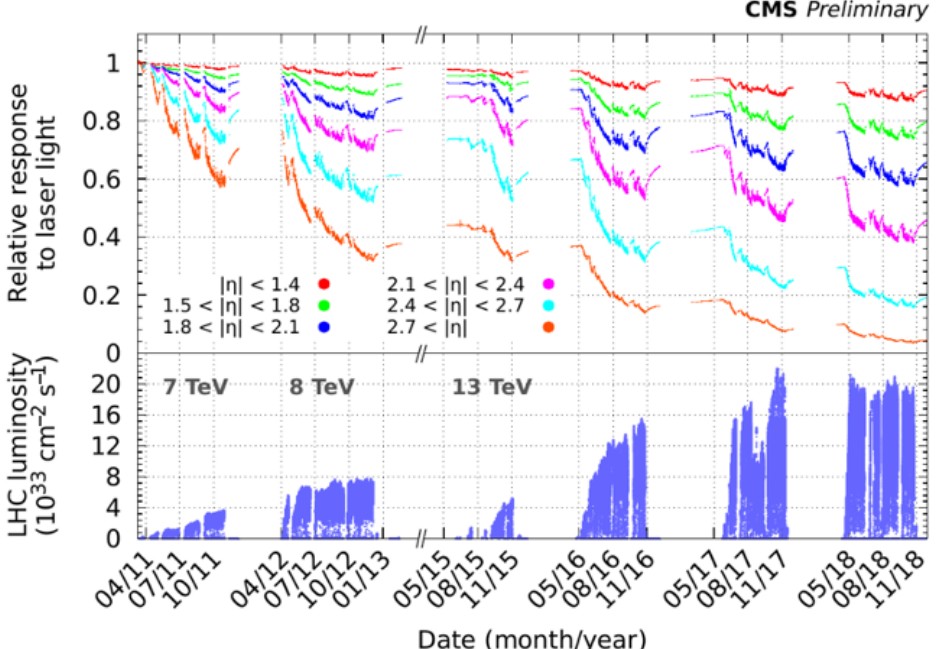

**Figure 3.** The relative response of the CMS ECAL to laser light and LHC instantaneous luminosity as a function of time plotted in different $\eta$ ranges [5].

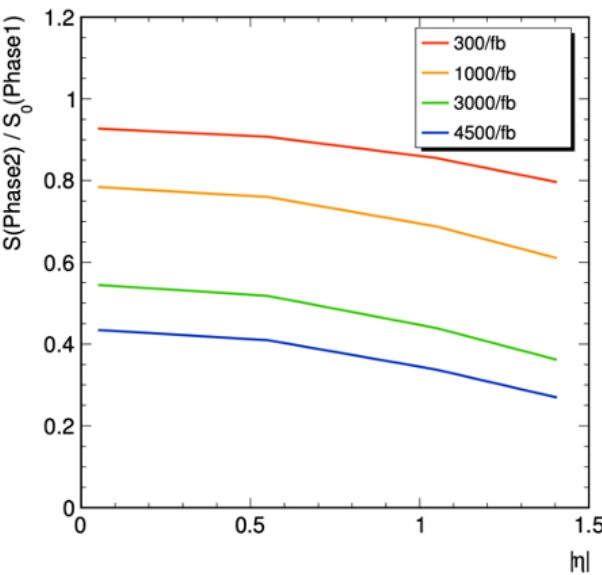

**Figure 4.** Predicted crystal light output relative to conditions at the start of CMS running, plotted as a function of of $\eta$ for various integrated luminosities for 50 GeV photon showers with HL-LHC electronics at 9 °C [6].

### 3.2. Avalanche Photodiode Longevity

There are two causes of radiation damage to APDs, gamma rays and hadrons. Gamma rays create surface defects that increase the surface current and reduce the quantum efficiency. Hadrons create bulk damage that causes an increase in the bulk current. The main concern for the HL-LHC is the increased dark current, as the electronic noise depends on the square root of bulk current. It can be mitigated by reducing the operating temperature, as shown in Figure 5.

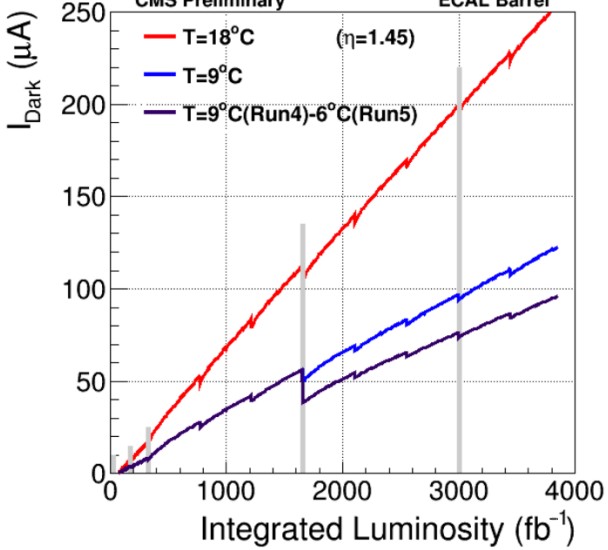

**Figure 5.** Dark current for different operating conditions. The current for APDs operated at 18 °C is shown in red, while that for APDs at 9 °C is shown in blue. A third possibility is shown in purple, where the APDs are operated at 9 °C in Run 4 (the first data taking period of HL-LHC), with the temperature reduced to 6 °C in Run 5 (the second data taking period of HL-LHC, after a shutdown which allows for experimental upgrades). The vertical shaded lines indicate long shutdowns [6].

### 3.3. Spike Rejection

Spikes are large isolated signals due to hadron interactions within the APD volume. Figure 6 shows a spike recorded in CMS, which shows a large signal in one readout channel. The pulse shapes caused by spikes and electromagnetic showers can be seen in Figure 7. In the current system, the spikes and EM showers are similar in both the start time and duration of the pulse. However in the HL-LHC the upgraded electronics will provide more discrimination between the two, as the spikes will have a faster rise and a shorter pulse shape, which will allow spike signals to be rejected. Spikes will dominate the L1 trigger rate at the HL-LHC if not suppressed. The expected event rates for the HL-LHC with 200 pileup events per bunch crossing can be seen in Figure 8. The ECAL will implement a spike killing algorithm in an off-detector FPGA and send this information as part of the trigger primitives to achieve >99.9% spike rejection in the L1 trigger. This level of spike rejection will leave less than 1 kHz residual spikes to trigger L1 for signals with $E_T > 5$ GeV.

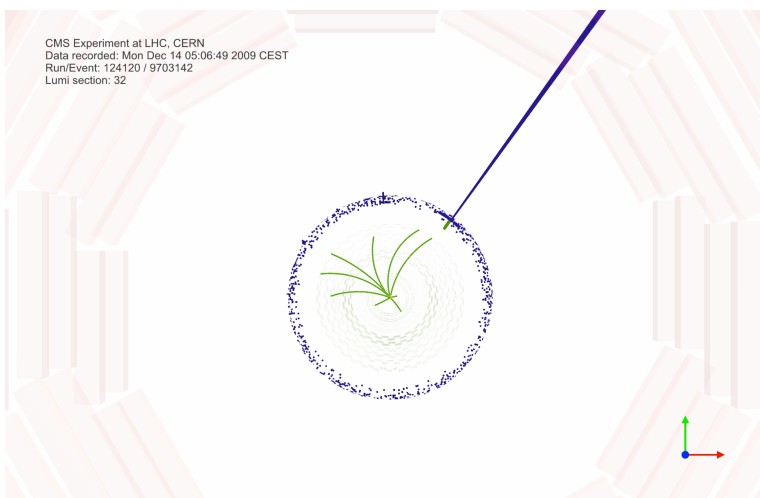

**Figure 6.** Event display showing a spike signal where a large amount of energy is deposited in a single ECAL readout channel. Tracks are shown in green and the energy deposited in the ECAL is shown in blue, with the spike event at approximately 1 o'clock.

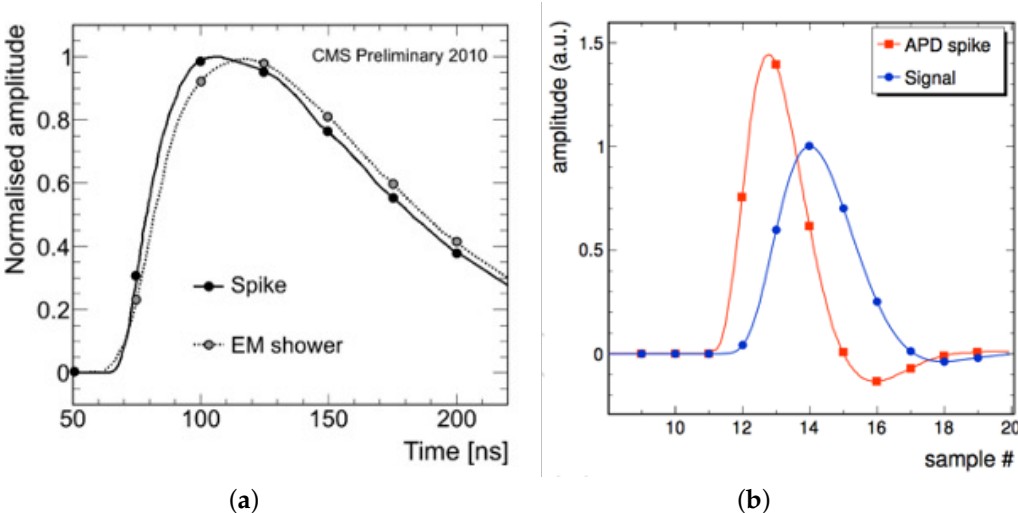

(**a**)          (**b**)

**Figure 7.** (**a**) Representative spike and EM shower pulse shapes in the current CMS ECAL, with the spike shown in black and the EM shower shown in grey. (**b**) Spike and EM shower pulse shapes in the CMS ECAL during the HL-LHC, with the spike pulse shown in red and the EM pulse shown in blue. HL-LHC pulse shapes are from the measured response of early prototype electronics convoluted with the known APD and scintillation pulse shapes.

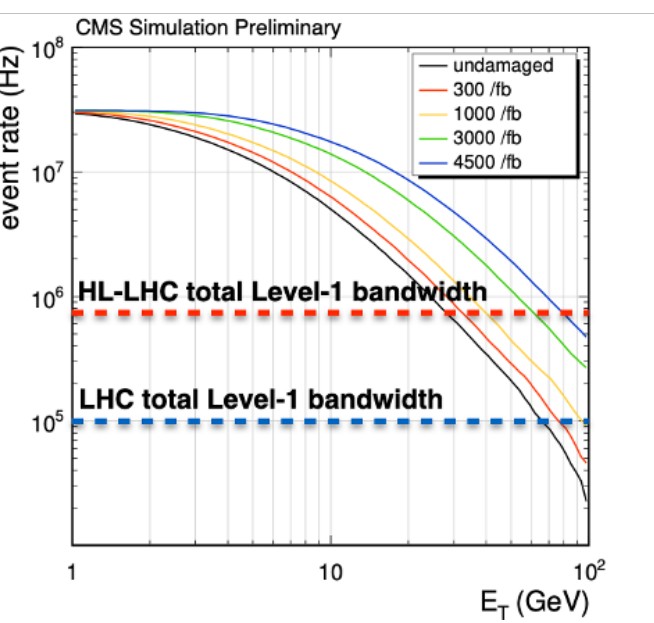

**Figure 8.** Expected rate of events for the HL-LHC in the CMS ECAL with 200 pileup, showing the energy deposits above specific thresholds after an integrated luminosity of 300, 1000, 3000 and 4500 fb$^{-1}$ without any spike suppression applied. The L1 bandwidth for the current LHC and the HL-LHC are also shown on this plot [6].

*3.4. Impact of Precision Timing*

It will be challenging to maintain reconstruction performance with 140–200 pileup events per bunch crossing. For the decay channel $H \to \gamma\gamma$, the primary vertex efficiency will be reduced from 75% to 30% in the absence of detector upgrades. However, improved vertex localisation is possible with precise (30 ps) timing capabilities. This precise timing gives a sensitivity gain of about 10% on $H \to \gamma\gamma$ resolution and fiducial cross-section relative to no precise timing case with 140 pileup events. With precision timing (ECAL + MIP timing) we can obtain similar results to Run 2 conditions, as shown in Figure 9. MIP timing will be provided by a new MIP timing detector, details of which can be found in [7].

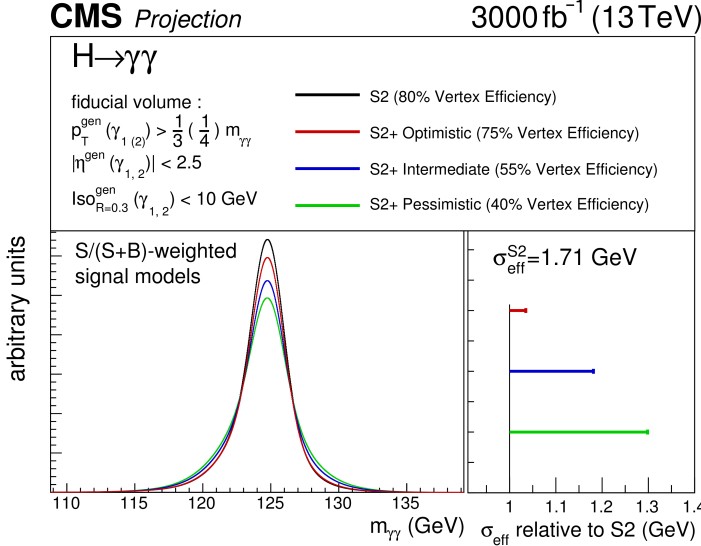

**Figure 9.** Lineshape for the H$\to \gamma\gamma$ signal in the four scenarios: no precise timing (green), precise timing in calorimeter (blue), precise timing in calorimeter and MIP timing (red) and Run 2 conditions (black) [8].

### 3.5. HL-LHC Electronics

As shown in Figure 2, the VFE in the HL-LHC readout system comprises of two ASICS namely the CATIA and LiTE-DTU. The CATIA is a pre-amplifier ASIC, using a trans-impedance amplifier (TIA) architecture with minimal pulse shaping. Faster pulse shaping is important for precise timing and improved spike rejection capabilities. It has two output gain values: ×1 and ×10. The CATIA V1 has been used in test beams with good results, and initial tests of CATIA V2.1 (the latest version) also show very promising results.

The LiTE-DTU is a data conversion, compression and transmission ASIC. It uses two 12-bit ADCs with 160 MS/s data conversion. It also has lossless data compression, a look-ahead algorithm where a sample saturation check prevents samples from different gains in the same APD signal timeframe being mixed. An effective number of bits (ENOB) scan has been performed as function of frequency with the results shown in Figure 10. These results match the expected ADC performance. In February, we received approximately 600 packaged LiTE-DTU v2. All 600 of these have been tested, with 98% passing. We have also performed a single event upset (SEU) test at CRC Louvain with no I2C errors and no PLL loss of lock.

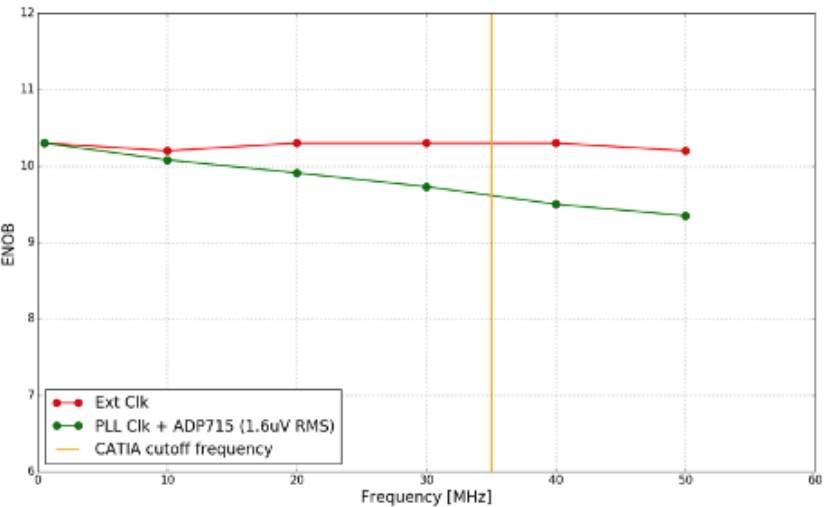

**Figure 10.** ADC ENOB as a function of the input frequency. The red data points show the ENOB using an external clock, while the green data points show the ENOB using a PLL clock and ADP715 (1.6 V RMS). The yellow line shows the CATIA cut off frequency.

We have also completed combined tests using both the CATIA v2.0 and the LiTE-DTU v2.0, using both single channel and multi channel test boards. Preliminary results show that the system is working well, with initial noise and timing measurements compatible with 30 ps timing for $E > 50$ GeV. The new LiTE-DTU features were also shown to be working. These features include test-pulse generation for CATIA, a secondary calibration scheme and ADC calibration with CATIA baseline-like levels. We are close to starting production for both ASICs.

The very front end (VFE) contains 5 × CATIA and 5 × LiTE DTU chips. It is responsible for the calibration of ADC values using CATIA's reference voltage and embedded multiplexer, as well as readout of temperature sensors (via FE) for components including APDs, CATIAs and PCBs. A pilot run of 8 VFE v3 boards has been tested, with a larger production launched at the beginning of May. The low voltage regulator (LVR) supplies voltages to the ASICS and FE board. It is a radiation tolerant ASIC that will readout APD leakage current, using DCDC converters (specifically bPOL12s [9] and linPOL12). The design is magnetic field tolerant and has low noise. The current versions of the VFE and LVR are tentatively the final versions.

The front end (FE) allows streaming of full granularity data off-detector at 40 MHz, which is not possible in the current detector. It sends a clock to the VFE directly from the

lpGBT controller and uses I2C via a controller-responder chain. The FE also monitors the APD dark current. The FE v3 is close to the final version.

The backend electronics is comprised of a barrel calorimeter processor (BCP) board, which combines trigger and DAQ functionality and provides clock and control signals to the FE electronics. Each board handles signals from 600 crystals and uses commercially available FPGAs. Algorithms are being developed using high level synthesis to produce trigger primitives. The BCP v1 uses one KU115 FPGA and is currently being used for integration tests with VFE/FE boards and DAQ. However, the BCPv2 will use one VU13P instead of two KU115's. This will provide the BCP with nearly three times the memory, 30% more logic cells and 11% more digital signal processing. Schematics for the BCP v2 are currently under development. Testing results for the BCP v1 (including timing performance) are good.

*3.6. Test Beam Results*

In 2021, we tested a single ECAL tower (5 × 5 crystal matrix) with prototype HL-LHC electronics, specifically the CATIA v1.2 and LiTE DTU v1.2, at the CERN SPS H4 beamline using an electron beam with energies from 25 to 250 GeV. The test beam setup is shown in Figure 11. The time resolution measurement is performed by comparing the time measured by a single channel to that of an external timing reference detector placed along the beamline, with the results shown in Figure 12. The constant term of about 12 ps meets the requirements for the HL-LHC design. A timing resolution of 30 ps for E > 50 GeV is shown to be achievable. Figure 13 shows the energy resolution using test beam data from 2018, where we used CATIA v1 and a commercial 160 MHz ADC. The setup was otherwise identical to that used in 2021. The energy resolution meets the HL-LHC requirements, with a constant term of <0.5%.

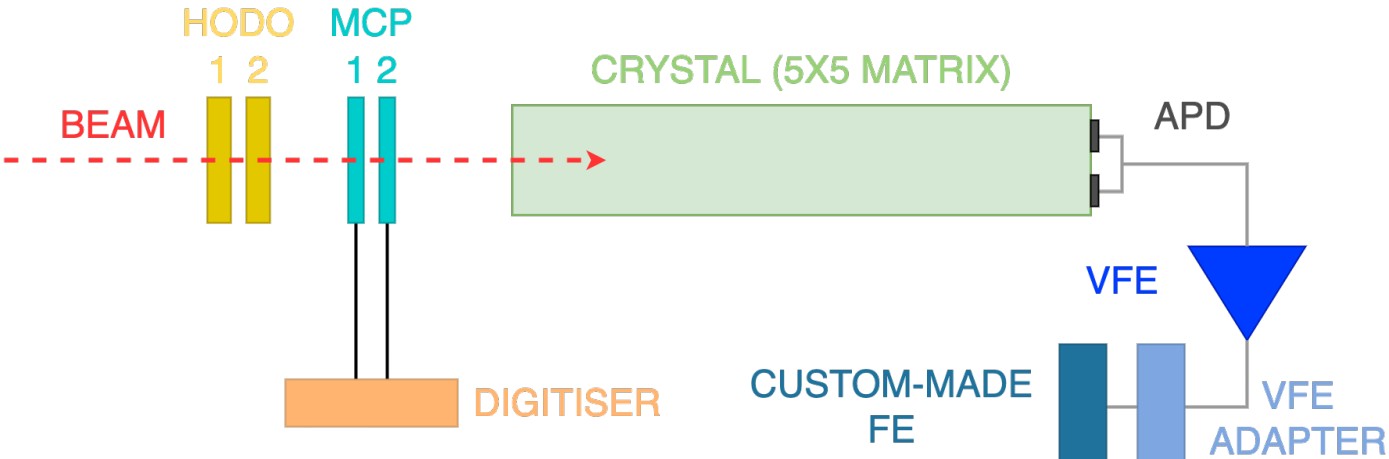

**Figure 11.** Test beam setup. Hodoscopes (hodo) were used to measure the beam position, while microchannel plate detectors (MCP) were used for external timing measurements. In 2018 the VFE was a commercial ADC, while in 2021 the LiTe-DTU was used.

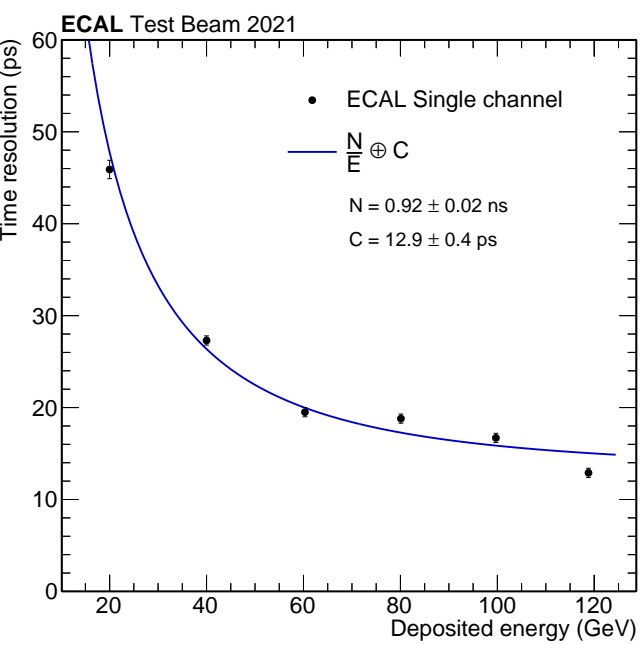

**Figure 12.** Time resolution as function of the deposited energy obtained in 2021 test beam data. The solid blue line represents the fit with the timing resolution function $N/E \oplus C$, where N denotes the noise, C the constant term, and E the deposited energy.

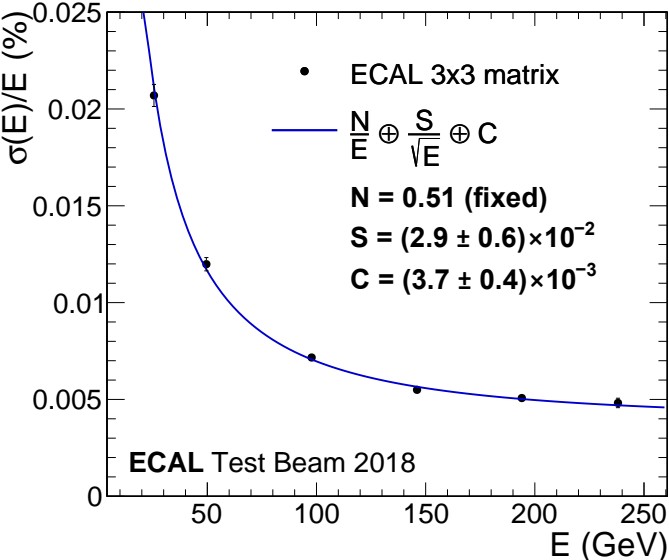

**Figure 13.** Energy resolution of a $3 \times 3$ crystal matrix as a function of the deposited energy obtained in 2018 test beam data. The solid blue line represents the fit with the energy resolution function $N/E \oplus S/\sqrt{E} \oplus C$, where N denotes the noise, S the stochastic term, C the constant term, and E the deposited energy.

## 4. Conclusions

Both the on- and off-detector electronics will be replaced in the barrel region of the CMS ECAL for the HL-LHC to maintain the current performance. Full featured ASICs have been received and initial tests show good results. The plan for this year includes a full system test of around 400 channels using a spare barrel supermodule. Towards the end of the year we will use this supermodule with the prototype HL-LHC electronics in a test beam with electrons and pions. We will also have an engineering design review to provide

a green-light for the production of the front-end ASICs and electronic boards. Development of the BCPv2 and associated firmware will continue. We will have production ready (and tested) versions of the ASICs and on-detector boards by the end of the year.

All 36 ECAL barrel supermodules will be extracted at the start of LS3. The current electronics will be exchanged for their HL-LHC counterparts and associated services by dedicated teams on the surface, with every supermodule being fully tested before reinstallation. This whole procedure will take approximately one year.

All barrel calorimeter components are on track for installation during LS3.

**Funding:** The author is supported by the Science & Technology Facilities Council (STFC). This work was done within the context of the CMS Collaboration.

**Institutional Review Board Statement:** Not applicable.

**Informed Consent Statement:** Not applicable.

**Data Availability Statement:** Not applicable.

**Conflicts of Interest:** The author declares no conflict of interest.

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
