# Peer review of "Upgrade of the CMS Barrel Electromagnetic Calorimeter for the High Luminosity LHC"

_instruments, doi:10.3390/instruments6030029_

Round 1

Reviewer 1 Report

The paper provides important information on the upgrade of the barrel ECAL of CMS, which is a crucial part of the necessary upgrades for HL-LHC. However, there are a few minor and also some more relevant shortcomings, which should be remedied before publication. I will list my comments below.

- line 40: "on detector electronics" --> "on-detector electronics"

- Figure 2: The graphics extend beyond the margin of the main text - I would expect that that is not tolerable for the published version. However, just reducing the size of the graphics to fit would likely make some of the details (in particular the text in the figure) too small. I would suggest to rework the figure so that it fits while keeping details legible.

in the caption:  "CMS ECA" --> "CMS ECAL"

- line 68: "can seen" --> "can be seen"

- Figure 4: For the interpretation of this figure it is important to understand, what the values are for the conditions up to (including) Run2. I suppose that the 300/fb results give those, but then they are for a temperature of 9°C. Please mention this either in the legend, caption, or corresponding text. Is the temperature difference to the current conditions negligible for this comparison? Is the value for 300/fb consistent with Figure 3, provided that those are at a different temperature?

- lines 82/83: You discuss here that you can achieve a better performance by a reduced operating temperature. However, nowhere do you even touch any integration issues of the upgrade. I understand that this is not the focus of the contribution, but, while one might easily consider that replacing electronics does not require a major effort on integration, this is not at all obvious for a cooling system. It would be important to make a statement about the implications for detector integration - can one really just use the existing cooling system to lower the temperature to 9°C, or even 6°C as you mention later? This question is also raised by the fact that you consider to lower the temperature in two steps. Why does one do that, if not because the cooling system has to be modified? If no modification is necessary, why do you not go immediately to 6°C?

- Figure 6: Details in the figure are not legible, and it is also not explained, what is shown. Please do that, preferably in the caption. 

- Figure 7: How is the plot on the right hand side obtained? From the caption I gather that it is not measured. Is this a simulation? Then say so. How realistic is this? Are there measurements using early version of the electronics?

- lines 89-96: In the estimates of the spike rates, have you taken into account that ageing effects will likely reduce the amplitude of physics signals more than that of the spikes, because the former will suffer additionally from a reduced light output of the crystal, so the relative amplitude of spikes will increase?

- Figure 8: The features of this figure are not well explained. From the discussion I would assume that the event rates are essentially spike rates here - you may mention that. How do you know that the spike rate goes down so dramatically with increased integrated luminosity? It would be very interesting to see in the same figure the physics event rates and the spike event rates together, so that one can see what level of spike rejection is needed.

- Lines 103/104 and Figure 9: Here you mention ECAL + MIP timing, but you do not explain it. Please do. In the figure, this is also marked as the optimistic scenario. If one assumes the intermediate (realistic?) scenario, the gain by the timing is not very large. If you can't get the optimistic scenario to work, is the intermediate one really worth the effort? This might be illustrated more strongly, if you could show the expected performance at HL-LHC without any improvement from timing.

- Figure 10: Here, the caption is very close to the adjacent main text. I would think this is not desirable, and must have been introduced forcibly - may be by a negative \vspace. For clarity of reading, there should be a reasonable gap between caption and text. 

Reviewer 2 Report

Line 12      'consistent'

Lines 29-32   A comparison with the previous results is suggested to evaluate the improvements

Fig. 2 caption:  'ECA' to 'ECAL barrel'

Line 60     the h coverage of barrel must be specified

Lines 70-72  Fig.4 refer to Montecarlo simulation and not to test beam data.

Line 72   'see' to 'seen'

Fig.4    'at 9 oC' to 'with electronics at 9 oC'

Line 94   About the 'killing algorithm' the conclusion is not supported by any simulation or data

Line 104  What is providing the 'MIP timing'?

Line 121 Add a space from fig. 10

Everywhere     Unspecified acronyms

Round 2

Reviewer 1 Report

The authors have adequately answered my questions and performed modifications where needed.
To confirm, I indeed made a mistake in my question 8, this should refer to the transverse energy dependence.
Regarding the figures, if the way they are is indeed accepted by the publisher, I have no objection.